# Ciclopirox Olamine Induces Proliferation Inhibition and Protective Autophagy in Hepatocellular Carcinoma

**DOI:** 10.3390/ph16010113

**Published:** 2023-01-12

**Authors:** Xinyan Wan, Junqi Xiang, Hui Fan, Ying Jiang, Yiting Lu, Chundong Zhang, Ying Zhang, Quanmei Chen, Yunlong Lei

**Affiliations:** Department of Biochemistry and Molecular Biology, Molecular Medicine and Cancer Research Center, College of Basic Medical Sciences, Chongqing Medical University, Chongqing 400016, China

**Keywords:** CPX, hepatocellular carcinoma, autophagy, ROS

## Abstract

Hepatocellular carcinoma is one of the most common fatal malignancies worldwide. Thus far, the hepatocellular carcinoma prognosis has been bleak due to deficiencies in the identification and diagnosis of early hepatocellular carcinoma. Ciclopirox olamine (CPX) is a synthetic antifungal agent and has been considered as an anti-cancer candidate drug recently, though the detailed mechanisms related to its anti-cancer effect in hepatocellular carcinoma have not yet been revealed. Here, we found that CPX could inhibit proliferation in HCC cells but not in intrahepatic cholangiocarcinoma cells by arresting the cell cycle. Moreover, the anti-cancer effects of CPX in HCC cells were also attributed to CPX-triggered ROS accumulation and DJ-1 downregulation. Additionally, CPX could promote complete autophagic flux, which alleviated the anti-cancer effect of CPX in HCC cells, whereas the ROS scavenger (NAC) would attenuate CPX-induced protective autophagy. Interestingly, CPX could also induce glycogen clustering in HCC cells. Altogether, this study provides a new insight into the detailed molecular mechanisms of CPX as an anti-cancer therapy and a strategy for treating hepatocellular carcinoma.

## 1. Introduction

As the fifth most common cancer worldwide, liver cancer is also the fourth leading cause of cancer-related deaths. There are two types of primary liver cancers, intrahepatic cholangiocarcinoma (ICC) and hepatocellular carcinoma (HCC), along with less common cancers such as clear giant cell carcinoma, sclerotic carcinoma and liver fibrolamellar carcinoma. ICC originates in the bile ducts, while HCC originates in the hepatocytes—the major parenchymal cells of the liver. HCC is a heterogeneous tumor with a high incidence, accounting for 80% of primary liver cancer cases worldwide [1,2,3,4]. Up to now, effective treatments for HCC have included hepatectomy, arterial chemoembolization, radiotherapy and so on [5,6]. It has also been reported that the use of natural compounds and nanotechnology can reduce systemic toxicity and side effects [7]. Even though modern treatments continue to advance, patients with HCC still have a poor prognosis due to its aggressive nature and rapid growth. Therefore, new targeted agents are imminently needed to enhance the therapeutic efficacy of HCC and improve the prognosis of patients.

Ciclopirox olamine (CPX) is a synthetic antifungal agent that has been widely used for decades to treat tinea corporis, tinea pedis, lichen planus and Candida albicans. Recent preclinical and clinical studies have suggested that CPX also has potential anti-tumor activity [8,9,10,11,12]. Several studies have shown that CPX inhibits proliferation and induces apoptosis in breast cancer by regulating the expression of cell cycle and apoptosis regulatory proteins [13,14,15]. In addition, our previous studies showed that the anti-colorectal and cervical cancer effects of CPX are achieved through the induction of apoptosis and the inhibition of cell proliferation by downregulating PARK7 and ROS accumulation [10,12]. Notably, clinical studies have shown good therapeutic indications and demonstrated the safety of CPX by assessing dose tolerance, pharmacokinetics and pharmacodynamics in patients with refractory malignant hematological diseases [16]. However, the specific molecular mechanism of CPX against hepatocellular carcinoma needs to be further investigated.

Autophagy is a conserved lysosomal degradation pathway, which is an orderly metabolic process. Long-lived proteins in the cell, damaged organelles and other accumulated goods will be swallowed by the double-membrane autophagic vesicles and eventually wrapped in the autolysosomes for degradation [17]. The role of autophagy in cancer is complex and is related to many biological factors, such as the tumor type, the stage of progression and the expression of tumor-related genes [18,19]. Relevant studies have shown that autophagy has a bidirectional effect on cancer cells that may be cell protective or cytotoxic. Numerous preclinical studies have shown that autophagy can be induced by a variety of targeted therapeutic agents and DNA damaging agents, and, in these studies, autophagy mostly plays a cytoprotective rather than cytotoxic role [20]. Therefore, it has been suggested that the modulation of autophagy can be used to enhance anti-tumor effects, and autophagy may be a useful developmental target and research area for clinical anti-tumor therapy. Glycogen metabolism is an important metabolic process in the liver that involves a variety of enzymes and proteins that can undergo structural, functional and expression-level changes during HCC development to achieve metabolic reprogramming. When the function of a glycogenolytic enzyme is impaired, it can induce glycogen storage diseases (GSDs), resulting in significant liver enlargement and the development of hepatocellular adenomas and carcinomas [4,21,22]. Glycogen accumulated in hepatocellular carcinoma has been reported to be able to drive tumor development by inhibiting the activity of the Hippo signaling pathway through liquid–liquid phase separation [23].

In this study, we found that CPX significantly inhibited the ability of HCC cells to proliferate and significantly impaired the cell cycle. In addition, CPX induced protective autophagy and the accumulation of ROS and decreased the expression of DJ-1. Notably, we found that glycogen clustering was observed in HCC cells treated with CPX. These findings provide new perspectives on clinical treatment strategies for HCC.

## 2. Results

### 2.1. CPX Inhibits Proliferation in HCC Cells

To validate the anti-cancer effect of CPX in liver cancer cells, CCK8 assay was performed to evaluate the viability of various liver cancer cell lines treated with different concentrations of CPX for 24 h. As shown in Figure 1A, CPX treatment significantly decreased the cell viability of several HCC cell lines (sk-Hep1, Huh7, Hep3B and Lm9) with relatively low IC50 values (Figure 1B), while ICC cell lines including RBE and HuccT1 cells were less sensitive to CPX. Thus, the HCC cell lines sk-Hep1 and Huh7 were chosen to further reveal the anti-cancer effects of CPX on HCC. Consistently, the proliferative ability of HCC cells was significantly inhibited after CPX treatment, as confirmed by reduced clone numbers (Figure 1C,D). Furthermore, the EdU positive cell rate was significantly decreased in HCC cells after CPX treatment (Figure 1E,F). In addition, we also treated sk-Hep1 and Huh7 with different concentrations of CPX for 48 h and found that CPX could more significantly inhibit the proliferative ability (Appendix A). Cell cycle analysis revealed that CPX treatment hindered the cell cycle progression of hepatocellular carcinoma cells, as evidenced by the increased percentage of G1 phase cells and the decreased percentage of G2/M and S phase cells (Figure 2A). Then, the cytotoxicity of the drug was detected by lactate dehydrogenase (LDH) assays. We found that CPX-treated caused weak cytotoxicity in HCC cells (Figure 2B). Meanwhile, Annexin V/PI staining by flow cytometry further confirmed that CPX slightly induced apoptosis in hepatocellular carcinoma cells (Figure 2C). Collectively, these results indicate that proliferation inhibition is implicated in CPX against hepatocellular carcinoma cells.

### 2.2. CPX Induces Autophagy in HCC Cells

Considering the important role of autophagy in tumor therapy, and our previous studies showing that autophagy is regulated by CPX in colorectal cancer and cervical cancer cells, here we explore whether autophagy is involved in CPX-treated hepatocellular carcinoma cells. As shown in Figure 3A, CPX treatment increased the turnover of LC3-I to lipidated LC3-II in a dose-dependent manner in sk-Hep1 and Huh7 cells. The CPX-induced accumulation of autophagic vesicles (LC3 puncta) further supported the autophagic phenotype (Figure 3B,C). In addition, double-membraned vacuoles were observed by transmission electron microscopy in the cytoplasm of the CPX-treated cells, which is characteristic of autophagosomes. Nevertheless, double-membraned autophagic vacuoles were barely present in the control cells (Figure 3D,E). In summary, these data suggest that CPX induces autophagy in HCC cells.

To confirm whether CPX induced complete autophagic flux in HCC cells, the protein levels of p62, the autophagic substrate known to all, were examined. As shown in Figure 3A, a decreased p62 level was observed in the CPX-treated cells, along with the elevated LC3-II levels, indicating the induction of complete autophagic flux. We also used autolysosome inhibitors (Baf A1, CQ (10 μM, common concentration), or E64D and Pep A) to further detect the integrity of the autophagic flux and found that the accumulation of LC3-II was further enhanced under a combinational treatment of CPX with autolysosome inhibitors (Figure 3F–H). Additionally, performing a tandem monomeric GFP-RFP-tagged LC3 assay, we found that red fluorescent autophagosomes were significantly increased in CPX-treated cells, and that this transformed into yellow under co-treatment with CQ (1 μM, low concentration) (Figure 3I,J). Altogether, these findings indicate that CPX induces complete autophagic flux in HCC cells.

### 2.3. Autophagy Attenuates CPX-Induced Anti-Cancer Effects

To confirm the role of autophagy in the anti-HCC effect of CPX, sk-Hep1 and Huh7 cells were treated with CPX in combination with the autophagy inhibitor CQ. Compared to CPX alone, co-treatment with CQ decreased the cell viability of hepatocellular carcinoma cells (Figure 4A). A marked decrease in cell proliferation was also observed, as evidenced by EdU labeling (Figure 4B,C). In conclusion, these results suggest that autophagy attenuates the anti-cancer effects of CPX.

### 2.4. Downregulation of DJ-1 Is Involved in the Anti-Cancer Effects of CPX

We have previously found that the expression of DJ-1 was notably reduced in CPX-treated colorectal cancer and cervical cancer cells [10,12]. We assumed that DJ-1 might be a target of CPX for tumor treatment. To confirm whether DJ-1 was involved in the CPX-treated hepatocellular carcinoma cells, the expression of DJ-1 was detected. CPX treatment led to the decrease in DJ-1 in a dose-dependent manner by transcriptional inhibition in sk-Hep1 and Huh7 cells (Figure 5A,B). In addition, we found that the exogenous expression of DJ-1 attenuates the proliferation inhibition induced by CPX in HCC cells (Figure 5C–E). However, the level of LC3-II remained unchanged with DJ-1 overexpression in HCC cells (data not shown). These data indicated that CPX could also inhibit the proliferation of HCC cells by targeting DJ-1, but that DJ-1 was not involved in CPX-induced autophagy.

### 2.5. ROS Are Responsible for the Anti-Cancer Effects of CPX

In our previous studies, we found that CPX could play an anti-cancer role by targeting DJ-1 through ROS accumulation. In addition, increasing evidence indicates that ROS can also play an important role in controlling autophagy [24]. To determine whether ROS was involved in the HCC cells treated with CPX, we detected the levels of intracellular ROS in sk-Hep1 and Huh7 cells in the presence or absence of CPX treatment. As shown as Figure 6A, CPX could trigger ROS accumulation in HCC cells. To investigate the role of ROS in the anti-cancer effects induced by CPX, Huh7 cells were treated with CPX combined with NAC, a ROS scavenger. Compared to CPX alone, the combinational use with NAC increased the viability of Huh7 cells (Figure 6B). A similar increase in cell proliferation was also observed as evidenced by EdU labeling (Figure 6C,D). In addition, the NAC treatment reduced the LC3B-II turnover induced by CPX (Figure 6E). These results suggest that ROS are responsible for the CPX-induced proliferation inhibition and autophagy.

We have found that ROS trigger glycogen clustering in CPX-treated cervical cancer cells. To determine whether the same phenomenon exists in hepatocellular carcinoma cells, we performed a PAS staining assay. We also found that CPX induced glycogen clustering in hepatoma cells in a dose-dependent manner (Appendix A), suggesting that glycogen clustering may play a role in the anti-hepatocellular carcinoma effect of CPX. Overall, these results suggest that ROS accumulation and glycogen clustering may play a role in the anti-hepatocellular carcinoma effect of CPX.

## 3. Discussion

CPX is an antifungal agent that has a killing effect on Onychomycosis caused by pathogenic *fungi*, and the efficacy and safety of CPX have been well documented [25,26]. Recent studies have shown that anti-cancer activity was shown by CPX [27,28,29]. The mechanisms of the anti-cancer effect of CPX are multiple and complex. In many cancers, chelation of intracellular iron is probably a vital molecular mechanism for CPX-induced cell death. CPX could inhibit the certain iron-dependent enzymes, such as ribonucleotide reductase, deoxyhypusine hydroxylase and prolyl 4-hydroxylase, as well as blocking the Wnt/β-catenin and mTORC1 signaling pathways through iron-related chelation, which will lead to proliferation inhibition and cell death in cancer cells [30,31]. In addition, CPX can serve as a canonical ferroptosis inhibitor and can inhibit ferroptosis-related cell death [32]. However, the molecular mechanisms of CPX in hepatocellular carcinoma cells have not been clearly determined. In the present study, we found that CPX showed remarkable anti-tumor effects on HCC cells, but not on ICC cells, by inducing proliferation inhibition, which was achieved through an arrested cell cycle.

Autophagy is activated to degrade misfolded proteins and damaged or senescent organelles in response to cellular starvation or stresses. The digested products are recycled to support cell metabolism and maintain energy homeostasis [33]. The function of autophagy in the treatment of cancer is complicated [34,35]. On one hand, autophagy may lead to cell death [36,37]; on the other hand, autophagy may play a supporting role in drug resistance [38]. In this study, we confirmed that CPX can promote the combination of autophagosome and lysosome to form autolysosome, which means that CPX promotes a complete autophagic flux in hepatocellular carcinoma cells. Interestingly, CPX-activated autophagy plays a cytoprotective role in hepatocellular carcinoma cells. The blockage of autophagy using CQ significantly enhances the anti-cancer effects of CPX. Therefore, autophagy may be a novel target to overcome the drug resistance of hepatocellular carcinoma.

DJ-1, a well-known oxidative stress sensor, is mainly located in the mitochondria and plays a cellular protective role by eliminating oxidative stress [39,40]. Multiple studies have shown that DJ-1 is overexpressed in a variety of tumors and is positively correlated with tumor progression, tumor recurrence and chemotherapy resistance [41,42]. CPX was found to inhibit the proliferative ability of colorectal and cervical cancer cells by targeting DJ-1 through inducing ROS accumulation in our previous studies [10,12]. In this study, CPX also decreased the expression of DJ-1 in a concentration-dependent manner, while DJ-1 overexpression would markedly impair the CPX-induced suppression of cell proliferation in HCC cells. Therefore, DJ-1, as an oncogene, may become a potential therapeutic target in HCC.

Amounts of drugs directly induce drastic increases in the intracellular ROS levels, which accounts for its cytotoxicity to cancer cells. It is generally accepted that therapeutic stimulation of ROS can be an effective strategy to preferentially eliminate cancer cells [43,44]. Interestingly, CPX induces ROS accumulation in HCC cells, and the presence of N-acetyl-L-cysteine (NAC), a ROS scavenger, attenuated CPX-induced proliferation inhibition and protective autophagy. In our previous studies, ROS was found to play a critical role in the regulation of glycogen clusters and glycophagy in the CPX treatment of cervical cancer cells [10]. Moreover, in the present study, PAS staining results also showed that CPX-induced glycogen clustering in a concentration-dependent manner in hepatocellular carcinoma cells. Cancer cells survive and grow by altering the storage and breakdown of glycogen according to the stages of tumor progression and the microenvironment [23]. Based on these findings, we speculate that glycogen clustering may play an important role in the anti-HCC effects of CPX, which requires further experimental exploration.

In summary, our findings indicate that CPX inhibits the proliferation of HCC cells by blocking the cell cycle, decreasing DJ-1 expression and enhancing ROS accumulation. At the same time, CPX induced cell protective autophagy dependent on ROS in HCC cells. In addition, CPX could also induce glycogen clustering in HCC cells. These results provide a new basis for the clinical treatment of hepatocellular carcinoma. In subsequent experiments, we will further assess whether CPX prodrugs have protective effect in in vivo HCC models and reveal the role of glycogen clustering in the anti-HCC effect of CPX.

## 4. Materials and Methods

### 4.1. Cell Culture

Human hepatocellular carcinoma cell lines sk-Hep1, Huh7, Hep3B, HuccT1, RBE and Lm9 were purchased from the American Type Culture Collection (ATCC). Cells were routinely maintained in DMEM (Gibco, Grand Island, NY, USA), MEM (Gibco) and RPMI (Gibco, Grand Island, NY, USA), medium supplemented with 10% of fetal bovine serum (Hyclone, Logan, UT, USA), penicillin (10^7^/L) and streptomycin (10 mg/L) in a humidified incubator containing 5% CO_2_ at 37 °C.

### 4.2. Antibodies and Reagents

The following primary antibodies were used: LC3 (Sigma Aldrich Co, St Louis, MO, USA), p62 (Cell Signaling Technology, Boston, MA, USA), β-actin (Cell Signaling Technology).

Ciclopirox olamine (1134030), E64D (E8640) and Pep A (P5318) were purchased from Sigma. Bafilomycin A1 (HY-100558) and CQ (HY-17589A) were purchased from MedChemExpress (Trenton, NJ, USA).

### 4.3. Cell Viability and Proliferation Assays

Cell viability was determined by CCK8 assay. In brief, HCC cells were inoculated in 96-well plates at a density of 4000–6000 cells. After treatment, CCK8 (Bimake, B34302, Houston, TX, USA) reagents were added and incubated for 1 h. The absorbance value was then measured at 450 nm.

Colony formation assays were performed to analyze the long-term effects of CPX on HCC cell proliferation. A density of 800–1000 cells after treatment were inoculated in twelve-well plates and the medium was changed every 3 days. Two weeks later, cells were immobilized with 4% paraformaldehyde (Sigma) for 30 min and stained with Crystal Violet for another 30 min, and then the colonies were cleaned three times and photographed.

Cell proliferations were detected using EdU assay. Cells grown on coverslips were treated with CPX for 24 h and stained with EdU (RiBoBio, Guangzhou, China) for 12 h. Then, the excess dye was washed according to the instructions of the manufacturer. Finally, nuclei were stained with DAPI for 10 min and observed using fluorescence microscopy (Leica, Frankfurt, Germany).

### 4.4. Cell Cycle and Apoptosis Analysis

After staining cells with propidium iodide, the cell cycle distribution was determined by flow cytometry. In short, floating and adherent cells were collected in centrifuge tubes, washed with pre-cold phosphate-buffered saline (PBS), and fixed with 70% ethanol. The cells were then treated with 50 μg/mL of RNase A and 50 μg/mL of propidium iodide for 30 min at room temperature. The stained cells were analyzed using FACScalibur flow cytometer (BD Biosciences, Franklin Lake, NJ, USA).

The cells were stained with Annexin V-FITC and propidium iodide according to the protocol of the Annexin V-FITC Apoptosis Detection Kit (Sigma). The ratio of apoptotic cells was measured and analyzed by flow cytometry.

### 4.5. Immunoblot

The cells were lysed with RIPA buffer (Beyotime, Shanghai, China) added with a protease inhibitor cocktail (Sigma), and then the protein lysates were centrifuged and boiled with loading buffer. Finally, all lysates were quantified using the BCA Protein Assay (Thermo Scientific, Waltham, MA, USA) and analyzed by SDS-PAGE.

### 4.6. Transmission Electron Microscopy

The autophagic vesicles were observed by transmission electron microscopy assay. After treatment with CPX for 24 h, the Huh7 cells were immobilized in glutaraldehyde (Sigma), and ultrathin sections were prepared by a sorvall MT5000 microtome. Then, the sections were treated with lead citrate and/or 1% uranyl acetate and observed by Philips EM420 electron microscopy.

### 4.7. Immunofluorescence

HCC cells grown on coverslips that were treated with CPX for 24 h were immobilized with 4% paraformaldehyde (Sigma) for 30 min and then cleaned three times with PBS. Fixed cells were permeabilized with 0.5% Triton 100 for 20 min and blocked with 1% BSA for 2 h at 37 °C. Then, the cells were incubated with primary antibodies for 12 h at 4 °C, followed by incubation with secondary antibodies (Thermo Scientific) at 37 °C for 1 h away from light. Finally, the nuclei were stained with DAPI for 10 min. For the detection of autophagy flux, the cells were transfected with GFP-RFP-LC3 for 24 h and then treated with CPX for another 24 h. Images were captured using a confocal microscopy (Leica).

### 4.8. Transfection

pCMV-myc-DJ-1 plasmids were obtained as described previously [12]. The plasmids were transfected into the indicated cells using Lipofectamine 3000 (Invitrogen, Carlsbad, CA, USA) according to the manufacturer’s instructions.

### 4.9. Quantitative RT-PCR (qRT-PCR)

Total intracellular RNA was extracted by Trizol reagent (Invitrogen), and reverse transcription was performed using Reverse Transcription PrimeScript 1st Stand cDNA Synthesis kit (TaKaRa, Otsu, Japan). Following the instructions of manufacture, qRT-PCRs were performed using the quantitative PCR reagents SYBR PremixEx TaqTM (TaKaRa). The fold-changes were analyzed using the 2^−ΔΔCt^ method, which uses the levels of GAPDH quantified with target genes to act as an internal control. The qRT-PCR primers for DJ-1 were as follows: Forward primer: 5′-GTGCAGTGTAGCCGTGATGT-3′, Reverse primer: 5′-CCTCCTGGAAGAACCACCAC-3′. The qRT-PCR primers for GAPDH were as follows: Forward primer: 5′-ACCTGACCTGCCGTCTAGAA-3′, Reverse primer: 5′-TCCACCACCCTGTTGCTGTA-3′.

### 4.10. Reactive Oxygen Species (ROS) Measurement

According to the manufacturer’s instructions, the cells were stained with 29, 79-dichlorofluorescein diacetate (DCFH-DA) (GENMED, GMS10016.2, Shanghai, China) to detect the intracellular ROS level. The DCFH-DA signal was examined by an FACScalibur flow cytometer (BD Biosciences).

### 4.11. Glycogen Staining

Glycogen was determined with a standardized periodic acid Schiff (PAS) staining technique (Solarbio, G1360, Beijing, China). In brief, sk-Hep1 and Huh7 cells were immobilized with PAS Fixative for 10–15 min and then cleaned with water. Fixed cells were treated with Oxidant for 15–20 min at room temperature and then washed with water. Next, the cells were bathed with Schiff’s reagent for 15 min at RT away from light. After rinsing with running water, Mayer Hematoxylin Staining Solution was used to counterstain the cells. Images were captured using a confocal microscopy (Leica).

### 4.12. Data Analysis and Statistics

Data were shown as means ± s.d. All experiments were performed at least three times. Statistical analysis was performed by GraphPad Prism 8.0 software. The two-tailed Student’s *t*-test was used to determine statistical differences between groups. Significance was designated as follows: *, *p* < 0.05, **, *p* < 0.01, ***, *p* < 0.001.

## Figures and Tables

**Figure 1 pharmaceuticals-16-00113-f001:**
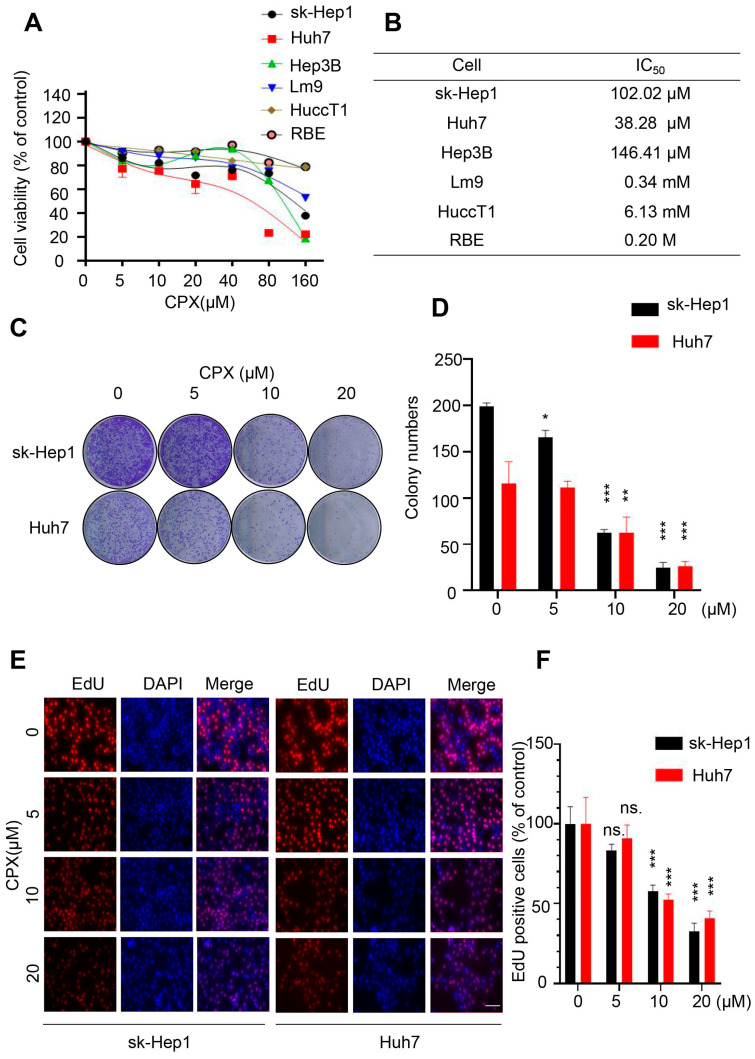
CPX inhibits proliferation in HCC cells. (**A**) The cell viability was determined by CCK8 assays. sk-Hep1, Huh7, Hep3B, HuccT1, RBE and Lm9 cells were treated with the indicated concentrations of CPX for 24 h. (**B**)The IC50 value of different cells treated with the indicated concentrations of CPX for 24 h. (**C**,**D**) Clone formation was performed to assess the cell proliferation rate. sk-Hep1 and Huh7 cells were treated with the indicated concentrations of CPX for 24 h; then, cells were inoculated into 12-well plates for two weeks and colony numbers were quantified. (**E**,**F**) EdU assay of HCC cells subjected to the indicated concentrations of CPX for 24 h. The EdU positive cell rate was quantitated. Scale bar, 100 μm. Data are means ± s.d. and are representative of 3 independent experiments. *, *p* < 0.05, **, *p* < 0.01, ***, *p* < 0.001. Statistical significance compared with respective control groups.

**Figure 2 pharmaceuticals-16-00113-f002:**
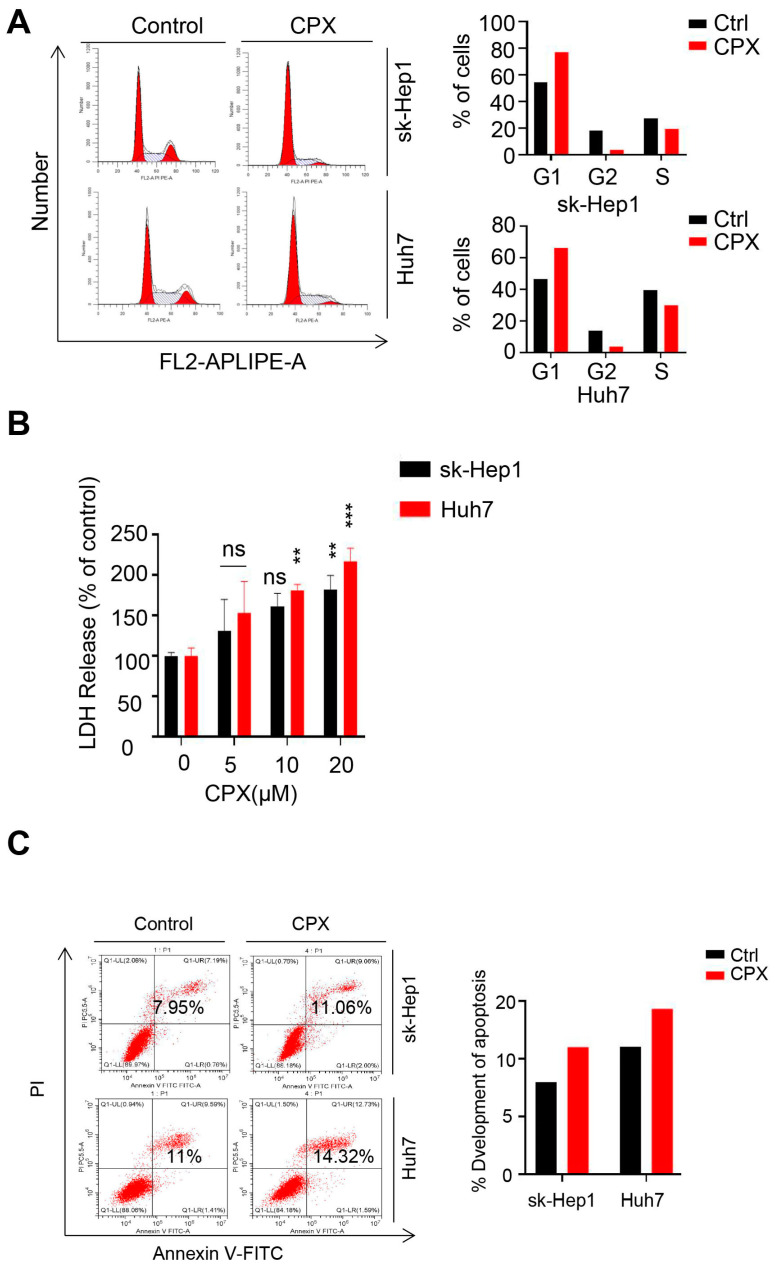
CPX hinders the cell cycle progression of HCC cells. (**A**). sk-Hep1 and Huh7 cells treated with or without 20 μM CPX for 24 h were subjected to flow cytometry analysis of the cell cycle distribution. (**B**) The release of dehydrogenase (LDH) in sk-Hep1 and Huh7 cells treated with indicated concentrations of CPX for 24 h was measured. (**C**) sk-Hep1 and Huh7 cells treated with or without 20 μM CPX for 24 h were stained with Annexin V/PI, and then detected by flow cytometry. The apoptosis rate was quantitated. Data are means ± s.d. and are representative of 3 independent experiments. **, *p* < 0.01, ***, *p* < 0.001. Statistical significance compared with respective control groups.

**Figure 3 pharmaceuticals-16-00113-f003:**
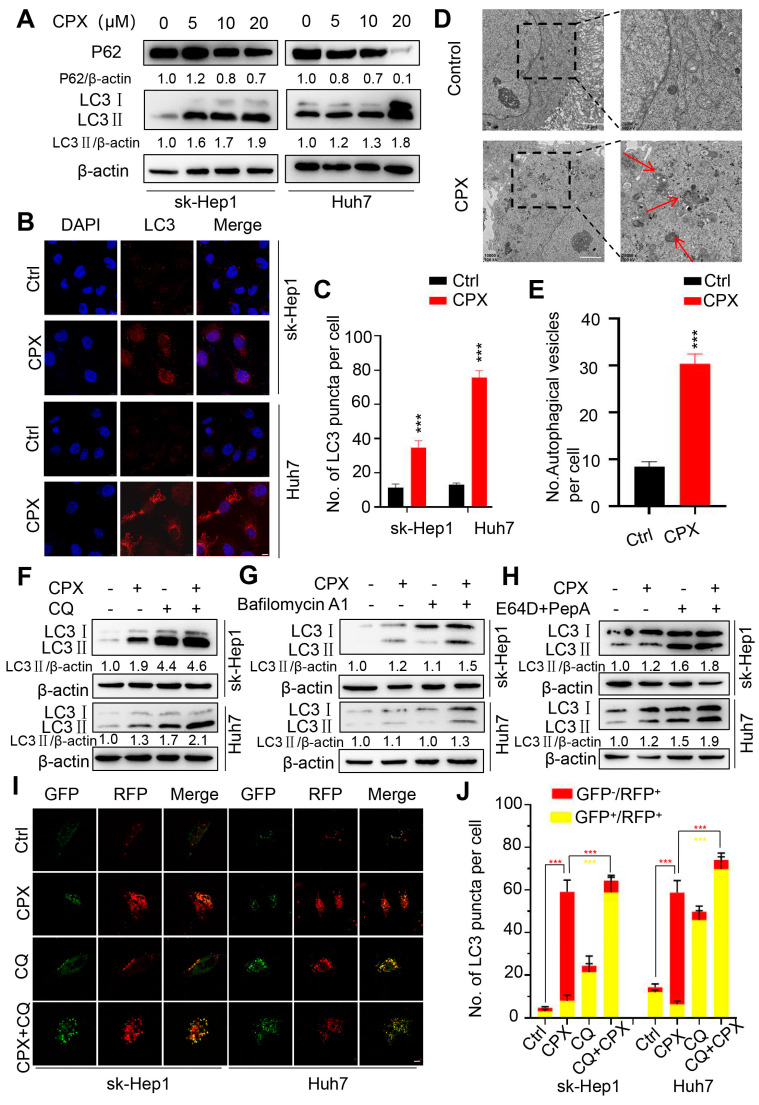
CPX induces autophagy in HCC cells. (**A**) Immunoblotting analysis of LC3 and p62 expression in sk-Hep1 and Huh7 cells treated with the indicated concentrations of CPX for 24 h. Relative intensity of LC3-II and P62 were quantified by normalization to β-actin by ImageJ software. (**B**,**C**) Immunofluorescence analysis of the formation of endogenous LC3B puncta in sk-Hep1 and Huh7 cells treated with or without 20 μM CPX for 24 h. Scale bar, 10 μm. (**D**,**E**) Transmission electron microscope was performed to observe the autophagic vesicles in Huh7 cells treated with or without 20 μM CPX for 24 h. Scale bar, 2 μm. (**F**) sk-Hep1 and Huh7 cells were treated with CPX (20 μM), CQ (10 μM) or a combination thereof for 24 h. The expression of LC3 was detected by immunoblot. Relative intensity of LC3-II was quantified by normalization to β-actin by ImageJ software. (**G**) sk-Hep1 and Huh7 cells were treated with CPX (20 μM) with or without Baf A1 (100 nM) for 24 h. The expression of LC3 was measured by immunoblotting. Relative intensity of LC3-II was quantified by normalization to β-actin by ImageJ software. (**H**) sk-Hep1 and Huh7 cells were treated with E64D (10 μg/mL) and Pep A (10 μg/mL) in the presence or absence of CPX (20 μM) for 24 h. The expression of LC3 was examined by immunoblotting. Relative intensity of LC3-II was quantified by normalization to β-actin by ImageJ software. (**I**,**J**) sk-Hep1 and Huh7 cells transiently transfected with tandem mRFP-GFP-tagged LC3 and treated with CPX (20 μM), CQ (1 μM) or a combination thereof for 24 h. Scale bar, 10 μm. GFP, GFP-LC3; RFP, RFP-LC3. The ratio of red puncta indicating autolysosome (GFP^−^/RFP^+^) versus yellow puncta indicating autophagosome (GFP^+^/RFP^+^) was quantitated. Data are means ± s.d. and are representative of 3 independent experiments. ***, *p* < 0.001. Statistical significance compared with respective control groups.

**Figure 4 pharmaceuticals-16-00113-f004:**
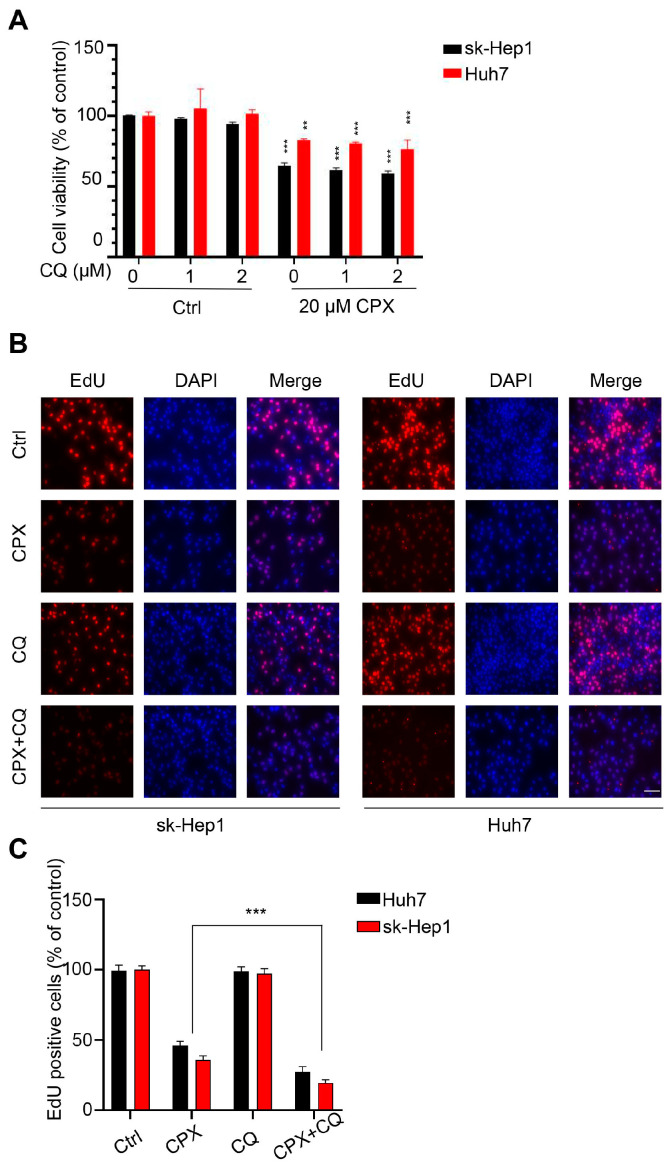
Autophagy attenuates CPX-induced anti-cancer effects. (**A**) The CCK8 assay detected cell viability of sk-Hep1 and Huh7 cells treated with the indicated concentrations of CQ with or without 20 μM CPX for 24 h. (**B**,**C**) The EdU assay detected proliferation of sk-Hep1 and Huh7 cells treated with 1μM CQ in the presence or absence of CPX (20 μM) for 24 h. The EdU positive cell rate was quantitated. Data are means ± s.d. and are representative of 3 independent experiments. **, *p* < 0.01, ***, *p* < 0.001. Statistical significance compared with respective control groups.

**Figure 5 pharmaceuticals-16-00113-f005:**
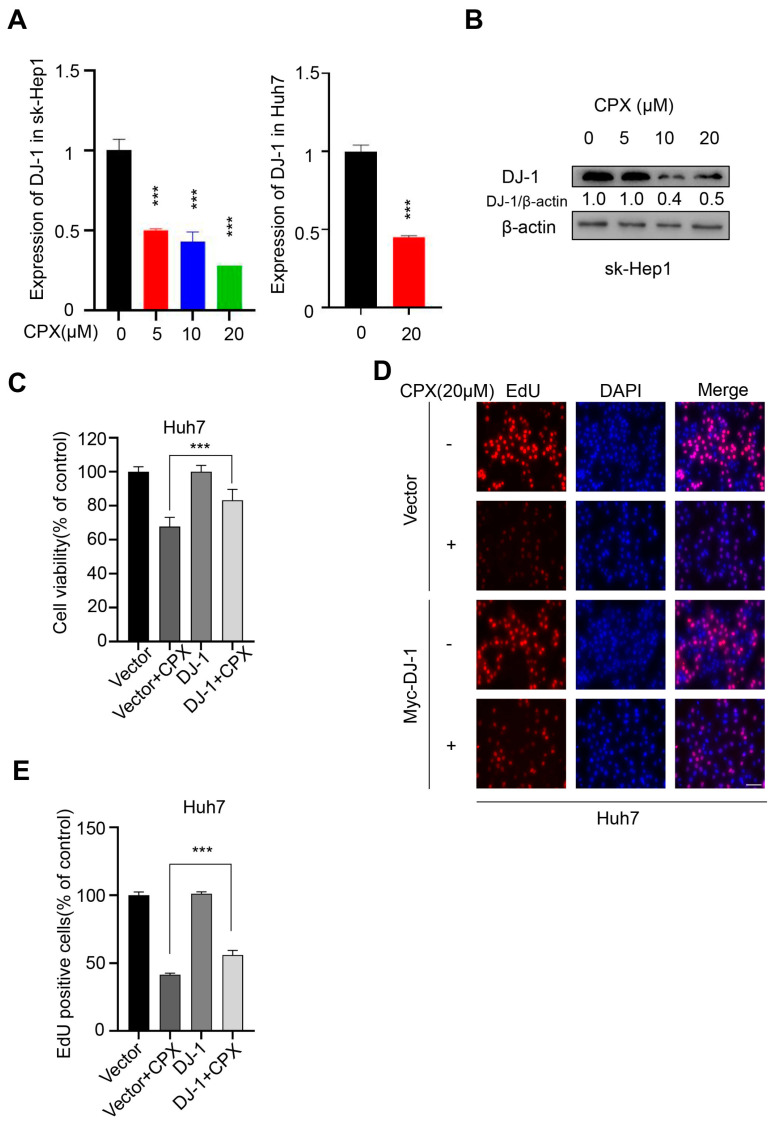
Downregulation of DJ-1 is involved in the anti-cancer effects of CPX. (**A**) The mRNA expression of DJ-1 in sk-Hep1 and Huh7 cells treated with indicated concentrations of CPX for 24 h was detected by qRT-PCR. (**B**) Immunoblotting analysis of DJ-1 expression in HCC cells treated with or without indicated concentrations of CPX for 24 h. Relative intensity of DJ-1 was quantified by normalization to β-actin by ImageJ software. (**C**) The cell viability was determined by CCK8 kit in HCC cells treated with or without 20 μM CPX for 24 h overexpressing DJ-1 and Vector. (**D**,**E**) The EdU assay detected proliferation of Huh7 cells treated with 20 μM CPX for 24 h overexpressing DJ-1 and Vector. The EdU positive cell rate was quantitated. Data are means ± s.d. and are representative of 3 independent experiments. ***, *p* < 0.001. Statistical significance compared with respective control groups.

**Figure 6 pharmaceuticals-16-00113-f006:**
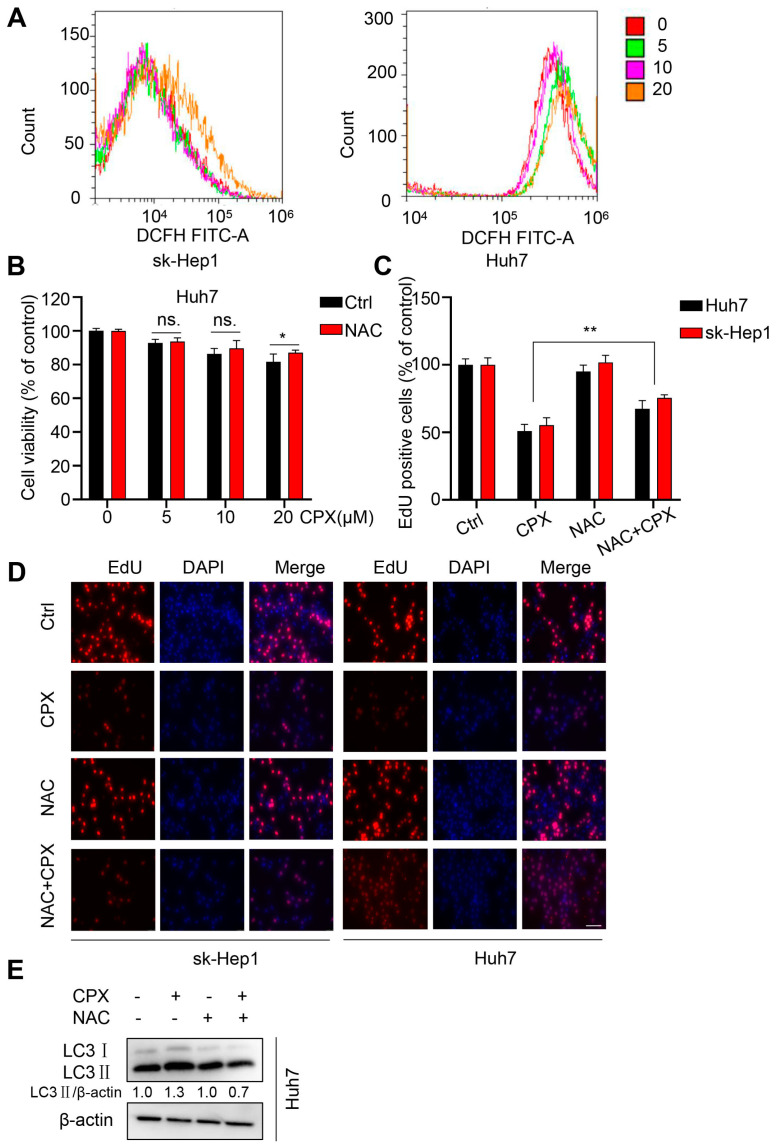
ROS are responsible for the anti-cancer effects of CPX. (**A**) ROS level was analyzed by DCFH-DA staining via flow cytometry in sk-Hep1 and Huh7 cells treated with indicated concentrations of CPX for 24 h. (**B**) The CCK8 assay determined cell viability of HCC cells treated with NAC (5 mM), CPX (20 μM) or a combination thereof for 24 h. (**C**,**D**) EdU assay of HCC cells treated with or without 5 mM NAC in the presence or absence of 20 μM CPX for 24 h. The EdU positive cell rate was quantitated. Scale bar, 100 μm. (**E**) Immunoblotting analysis of LC3 expression was detected in Huh7 cells treated with NAC (5 mM), CPX (20 μM) or a combination thereof for 24 h. Relative intensity of LC3-II was quantified by normalization to β-actin by ImageJ software. Data are means ± s.d. and are representative of 3 independent experiments. *, *p* < 0.05, **, *p* < 0.01. Statistical significance compared with respective control groups.

## Data Availability

Data is contained within the article and Appendix A.

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
