# Peer review of "Ciclopirox Olamine Induces Proliferation Inhibition and Protective Autophagy in Hepatocellular Carcinoma"

_pharmaceuticals, 2023, doi:10.3390/ph16010113_

Round 1
Reviewer 1 Report
In this manuscript, Xinyan Wan and colleagues describe the role of Ciclopirox olamine in hepatocellular carcinoma. The concepts in this manuscript are novel and relevant. However, a number of caveats needs to be addressed.
Specific comments:
1. The Edu staining is not convincing. Can only observe the decrease in fluorescence in CPX group. Should be show the Edu positive cell rate is decreased.
2. The quantification of Edu staining should be presented as Edu positive cell/total cell rate.
3. Rigor of this research is decreased by the absence of exact statistical information. It is necessary to perform statistical analysis on all western blot data.
4. The writing should be improved for a better understanding of the scientific content. For example: the complete autophagic flux in HCC cells could be induced by CPX, which could alleviate the anti-cancer effect of CPX.
5. CPX should replaced with Ciclopirox olamine in title.
6. In fig. 3C, what is LC3 puncta mean from TEM data.
7. Figure 3 and Figure 4 should be combined together.
8. Overall, the authors showed the phenotype of CPX on HCC. But there is absence of mechanism investigation study.
9. It is important to see whether CPX have protective effect in in vivo HCC model.
Author Response
1.The EdU staining is not convincing. Can only observe the decrease in fluorescence in CPX group. Should be show the EdU positive cell rate is decreased.
Response: Thanks very much for your valuable comments. The quantification data of EdU staining in figures was presented as EdU positive cell/total cell rate, which indicated as EdU positive cells (% control). However, we describe it as ‘EdU positive cell’ in Results and Figure legends. We have revised it as ‘EdU positive cell rate’ in the revised version. (Page 2, Line 88; Page 4, Line 109-110)
2.The quantification of Edu staining should be presented as Edu positive cell/total cell rate.
Response: Thanks very much for your valuable comments. The quantification data of EdU staining in figures was presented as EdU positive cell/total cell rate, which indicated as EdU positive cells (% control). However, we describe it as ‘EdU positive cell’ in Results and Figure legends. We have revised it as ‘EdU positive cell rate’ in the revised version. (Page 2, Line 88; Page 4, Line 109-110)
3.Rigor of this research is decreased by the absence of exact statistical information. It is necessary to perform statistical analysis on all western blot data.
Response: Thanks very much for your valuable suggestions. We have quantified all western blot data by normalization to control using ImageJ software.
4.The writing should be improved for a better understanding of the scientific content. For example: the complete autophagic flux in HCC cells could be induced by CPX, which could alleviate the anti-cancer effect of CPX.
Response: Thanks very much for your critical comment. We have checked English through the manuscript carefully during revision and invited a native English-speaking colleague to help us revise our English.
5.CPX should replaced with Ciclopirox olamine in title.
Response: Thanks very much for your valuable suggestions. We have replaced CPX with Ciclopirox olamine in title in the revised version. (Page 1, Line 2)
6.In fig. 3C, what is LC3 puncta mean from TEM data.
Response: Thanks very much for your valuable comments. First of all, I want to express my apologies. The quantification of TEM data should be indicated as ‘Number of autophagic vesicles per cell’. We have corrected the typo in the revised version. (Fig. 3E)
7.Figure 3 and Figure 4 should be combined together.
Response: Thank you very much for your valuable suggestions. We have combined Figure 3 and Figure 4 together in the revised version (Figure 3).
8.Overall, the authors showed the phenotype of CPX on HCC. But there is absence of mechanism investigation study.
Response: Thanks very much for your valuable comments. In the revised version, we further determined that the anti-cancer effects of CPX in HCC cells was also attributed to CPX-triggered ROS accumulation and DJ-1 downregulation. Additionally, CPX could promote complete autophagic flux, which alleviated the anti-cancer effect of CPX in HCC cells. Whereas the ROS scavenger (NAC) would attenuate CPX-induced protective autophagy. (Fig.5-6; Page 11-14, Line 194-249)
9.It is important to see whether CPX have protective effect in in vivo HCC model.
Response: Thank you very much for your kind comments. In this study, we mainly focus on revealing the role and mechanisms of CPX against HCC in vitro. Preclinical and clinical data has demonstrated that CPX possesses anticancer activity against a spectrum of human tumors, including leukemia, lymphoma, myeloma, Ewing’s sarcoma, colon adenocarcinoma, cervical carcinoma, renal cell carcinoma, esophageal cancer, bladder carcinoma, rhabdomyosarcoma, breast carcinoma, and pancreatic ductal adenocarcinoma (1). However, pharmacokinetic and pharmacodynamic studies showed that orally administered CPX was rapidly absorbed and cleared with a short half-life (Trial registration ID: NCT00990587). To reposition CPX for cancer therapy, efforts have been made to synthesize CPX derivatives and pro-drugs with high water solubility and bioavailability (1). In following study, we will assess whether CPX prodrug have protective effect in in vivo HCC model.
1.Huang Z, Huang S. Reposition of the Fungicide Ciclopirox for Cancer Treatment. Recent patents on anti-cancer drug discovery. 2021; 16: 122-35.

Reviewer 2 Report
In this study, Wan et al found that CPX could inhibit the proliferation in HCC cells but not in intrahepatic cholangiocarcinoma cells by arresting cell cycle. In addition, they also found that CPX could trigger protective autophagy, ROS accumulation and glycogen clustering in HCC cells.These findings are interesting. However, there are several comments and concerns that need to be addressed before the paper can be considered for publication.
1. In Fig.4, more experiments should be employed to determine the role of autophagy in CPX against HCC such as EDU assays.
2. The authors should detect the ROS level after treated with CPX in Huh7 cells.
3. The authors should reveal the role of ROS in CPX-induced proliferation inhibition in HCC cells.
4. The authors should reveal the role of ROS in CPX-induced autophagy in HCC cells.
5. The authors should examine whether DJ-1 was involved in CPX-induced proliferation inhibition and autophagy in HCC cells.
Author Response
1.In Fig.4, more experiments should be employed to determine the role of autophagy in CPX against HCC such as EdU assays.
Response: Thanks very much for your valuable comments. We further evaluate the role of autophagy in CPX against HCC cells using EdU assays. Markedly decrease in cell proliferation was observed in CPX-treated HCC cells in combination with CQ as evidenced by EdU labeling. (Fig. 4B-C; Page 10, Line 183-184)
2.The authors should detect the ROS level after treated with CPX in Huh7 cells.
Response: Thank you very much for your valuable suggestions. We have detected the ROS level after treated with CPX in Huh7 cells, and found that could trigger ROS accumulation in Huh7 cells. (Fig. 6A; Page 13, Line 221)
3.The authors should reveal the role of ROS in CPX-induced proliferation inhibition in HCC cells.
Response: Thank you very much for your valuable suggestions. We have assessed the role of ROS in CPX-induced growth suppression and found that combinatorial treatment of CPX with ROS scavenger N-Acetyl-cysteine (NAC) attenuated CPX-induced growth suppression, as evidenced by increased EdU incorporation and cell viability in combinatorial treatment cells, indicating that ROS are responsible for CPX-induced proliferation inhibition. (Fig. 6B-D; Page 13, Line 223-227)
4.The authors should reveal the role of ROS in CPX-induced autophagy in HCC cells.
Response: Thank you very much for your valuable suggestions. We have assessed the role of ROS in CPX-induced autophagy and found that combinatorial treatment of CPX with ROS scavenger (NAC) resulted in reduced LC3-II turnover, indicating that ROS are involved in CPX-induced autophagy. (Fig. 6E; Page 13, Line 227-229)
5.The authors should examine whether DJ-1 was involved in CPX-induced proliferation inhibition and autophagy in HCC cells.
Response: Thank you very much for your valuable suggestions. We have examined the role of DJ-1 in CPX-induced proliferation inhibition and autophagy, and found that exogenous expression of DJ-1 attenuates the proliferation inhibition induced by CPX in HCC cells, but DJ-1 wasn’t involved in CPX-induced autophagy (Fig. 5; Page 12, Line 200-204)

Reviewer 3 Report
The manuscript describes the effects of the antifungal drug ciclopirox on hepatocellular carcinoma cells. Repurposing antifungal drugs against cancer appears to be a promising strategy to find new cancer treatments. The presented results are interesting, but there are some points I request the authors to deal with in a revised version of the manuscript:
Introduction: ´´tinea pedis´´ occurs twice. Please correct. Write scientific names of fungi in italics.
Results: Please add IC50 values from the CCK8 assays. In addition, 24 h appears to be a relatively short incubation time for CPX. I suggest a longer incubation time to obtain reasonable results.
Please specify ´´CQ´´ for readers who are not familiar with this abbreviation.
Figures 1A and 4E: Please correct ´´vibility´´.
Figure 4E: Please explain ´´puntca/cell´´ in the caption.
Discussion: CPX is a hydroxamic acid derivative. Please discuss interactions of ciclopirox with vital metal ions as possible drug mechanisms in the light of the described results and mechanisms for this drug in HCC.
Author Response
1.Introduction: ´´tinea pedis´´ occurs twice. Please correct. Write scientific names of fungi in italics.
Response: Thank you for pointing out this mistake. We have corrected the typo in the revised version. (Page 1, Line 42; Page 15, Line 261)
2.Results: Please add IC50 values from the CCK8 assays. In addition, 24 h appears to be a relatively short incubation time for CPX. I suggest a longer incubation time to obtain reasonable results.
Response: Thanks very much for your valuable comments. We have added the IC50 values from the CCK8 assays in Fig 1. The IC50 value of different cells treated with the indicated concentrations of CPX for 24 hours. In addition, we also treated sk-Hep1 and Huh7 with different concentrations of CPX for 48h, and found that CPX could more significantly inhibited the proliferative ability in HCC cells (Fig. 1B and Fig. S1; Page 2, Line 89-91)
3.Please specify ´´CQ´´ for readers who are not familiar with this abbreviation.
Response: Thank you very much for your valuable suggestions. We have listed all abbreviation in the revised manuscript. (Page 1, Line 22-23)
4.Figures 1A and 4E: Please correct ´´vibility´´.
Response: Thanks. We have corrected this mistake.
5.Figure 4E: Please explain ´´puntca/cell´´ in the caption.
Response: Thanks very much for your valuable comments. First of all, I want to express my apologies. ´puntca/cell´ should corrected as ‘puncta per cell’. We have corrected the typo in the revised version. (Fig. 3J)
6.Discussion: CPX is a hydroxamic acid derivative. Please discuss interactions of ciclopirox with vital metal ions as possible drug mechanisms in the light of the described results and mechanisms for this drug in HCC.
Response: Thanks very much for your valuable comments. In many cancers, chelation of intracellular iron is probably a vital molecular mechanism for CPX-induced cell death (1,2). CPX could inhibit the certain iron-dependent enzymes such as ribonucleotide reductase, deoxyhypusine hydroxylase, and prolyl 4-hydroxylase, as well as block Wnt/β-catenin and mTORC1 signaling pathways by iron-related chelation, which will lead to proliferation inhibition and cell death in cancer cells (1,2). In addition, CPX can serve as a canonical ferroptosis inhibitor and can inhibit ferroptosis-related cell death (3). We have added this discussion in the revised version. (Page 15, Line 263-269)
1.Huang Z, Huang S. Reposition of the Fungicide Ciclopirox for Cancer Treatment. Recent patents on anti-cancer drug discovery. 2021; 16: 122-35.
2.Eberhard Y, McDermott SP, Wang X, Gronda M, Venugopal A, Wood TE, et al. Chelation of intracellular iron with the antifungal agent ciclopirox olamine induces cell death in leukemia and myeloma cells. Blood. 2009; 114: 3064-73.
3.Aguilera A, Berdun F, Bartoli C, Steelheart C, Alegre M, Bayir H, et al. C-ferroptosis is an iron-dependent form of regulated cell death in cyanobacteria. The Journal of cell biology. 2022; 221.

Round 2
Reviewer 1 Report
Authors addressed the comments of this referee.
Reviewer 2 Report
The authors have addressed all my concerns.
Reviewer 3 Report
The authors have dealt with my comments in an accurate way. I recommend acceptance of the revised manuscript.